# A multi-dimensional measure of pro-environmental behavior for use across populations with varying levels of environmental involvement in the United States

**Timothy J. Mateer** [1]*, **Theresa N. Melton**[1], **Zachary D. Miller**[2], **Ben Lawhon**[3], **Jennifer P. Agans**[1], **B. Derrick Taff**[1]

**1** Department of Recreation, Park, and Tourism Management, The Pennsylvania State University, University Park, Philadelphia, United States of America, **2** Intermountain Region-National Park Service, Philadelphia, Pennsylvania, United States of America, **3** Leave No Trace Center for Outdoor Ethics, Boulder, Colorado, United States of America

* tjm715@psu.edu

**Data Availability Statement:** All relevant data are within the paper and its Supporting information files.

## Abstract

Researchers continue to explore ways to understand and promote pro-environmental behavior (PEB) amongst various populations. Despite this shared goal, much debate exists on the operationalization and the dimensionality of PEB and how it is measured. This piece-meal approach to measurement has limited the ability to draw conclusions across studies. We address limitations associated with previous measures of PEB by developing a multi-dimensional scale that is validated across both a general population of individuals residing in the United States as well as a group of individuals associated with a pro-environmental organization. Exploratory and confirmatory factor analyses and reliability estimation were conducted for the developed measure across these two populations. Measurement invariance testing was also utilized to assess the psychometric stability of the scale across the two groups. Results indicated an 11 item scale was best fitting with two sub-scales: private and public behaviors. Implications for research and practice are discussed.

## 1. Introduction

As social-ecological systems continue to be plagued by multi-faceted, "wicked" environmental problems [1, 2], human behavior exists at the forefront of many of these environmental issues as well as their solutions [3, 4]. Given all human behavior impacts the natural world either directly or indirectly [5], environmental psychologists and environmental social psychologists are often at the forefront of efforts to understand and reduce the adverse ecological impacts associated with human behavior in various ways [6]. In linking human behavior to its environmental impacts, a variety of terms have been used in the academic literature including environmentally responsible behavior [7], environmentally significant behavior [5], general

**Funding:** The authors received support for this research from the Leave No Trace Center for Outdoor Ethics (https://lnt.org/). As one of the co-authors of this manuscript is employed by the supporting organization, the funder had a role in study design, data collection, and manuscript preparation.

**Competing interests:** The authors have declared that no competing interests exist.

ecological behavior [8], and pro-environmental behavior (PEB) [9, 10], the latter of which will be used throughout this study. As PEB becomes better understood and promoted, social-ecological systems can become more sustainable and resilient as well.

Measuring PEB effectively has significant implications for designing educational and psychological interventions aiming to encourage sustainable behavior within various populations [3, 11]. Be it an examination of barriers to acting in a pro-environmental manner [12] or formulating interventions to shift habitual behaviors towards more environmentally-friendly practices [13], environmental psychology has established itself as a major contributor to the conversation around behavioral change and environmentalism. While this body of research contributes important insight into how and why people behave in a pro-environmental manner, research on PEB has been broadly limited by inconsistencies in how it has been measured across studies. While some studies require researchers to measure specific, isolated behaviors [14, 15], many studies aim to understand how other social or psychological factors are related to more generalized PEB [16, 17]. The latter category is specifically constrained by many studies utilizing varied measures of general PEB. For example, [18] asserts that in a review of 49 studies measuring PEB through multi-item scales, 42 unique scales were utilized. As [19] further note, "many of these scales are ad hoc measures of unknown psychometric quality that have been developed for a particular research project" (p. 93). Such practices do not diminish the unique findings of these various studies, but the field of environmental psychology is limited in its understanding of broader trends due to this piecemeal approach to measurement.

In looking to other established scales in the field of environmental psychology, standardized measurements allow for findings to be more easily generalized for other psychological and behavioral constructs. For example, the New Ecological Paradigm represents a "gold standard" for measuring environmental attitudes that is broadly used across studies [20]. Similarly, Clayton's Environmental Identity scale [21] represents a general measure of environmental identity that is also utilized regularly (also see [22] for an updated environmental identity measure). Regarding measures of PEB specifically, [19] assert the General Ecological Behavior scale [8] "can probably be considered the best established of these domain-general propensity measures" (p. 93). However, this scale, while having advanced PEB research considerably in the past two decades, contains some measures which may be considered outdated or only tangentially linked to PEB (e.g., "If possible, I do not insist on my right of way and make the traffic stop before entering a crosswalk"). Given social and cultural perceptions of PEB have evolved in the two decades since the General Ecological Behavior scale was developed, updated measures of PEB have been created but display some limitations when being utilized in general populations. For example, some have been developed for specific sub-groups [23] or have displayed tenuous reliability and validity by categorizing behaviors by their environmental impact rather than their psychological properties [18].

Many scales measuring general PEB in the field of environmental psychology have been utilized in populations with a range of environmental orientations, values, and attitudes [16, 17, 24–26]. If a general PEB scale is to be effectively employed across these disparate groups, it should be developed and tested amongst the various populations within which it will ultimately be utilized [27]. With that in mind, this study aims to develop an updated measure of general PEB for both a general population of individuals living in the United States and a population associated with a pro-environmental organization (sampled from the Leave No Trace Center for Outdoor Ethics, to be discussed later). Furthermore, measurement invariance testing examines the psychometric stability of the developed scale across the two groups [28]. While we specifically develop this general PEB scale in the social, cultural, and infrastructural context of the United States, it may also potentially provide a basis for measuring general PEB in other countries as well.

## 2. Literature review

### 2.1 Understanding and measuring pro-environmental behavior

Understanding and promoting PEB has been a central topic in the field of environmental psychology [5, 9, 10, 19, 29]. While prior research focused heavily on these environmentally-focused behaviors, some debate exists in the field about what is and is not considered PEB [30]. For example, Stern [5] adopts an intention-oriented approach to understanding PEB, stating it is "defined from the actor's standpoint as behavior that is undertaken with the intention to change (normally, to benefit) the environment" (p. 408). Alternatively, Steg and Vlek [11] assert that PEB consists of actions taken that benefit or minimize the harm done to the natural environment, with the intention behind the behavior not being as heavily emphasized in their definition. As exemplified by Truelove and Gillis [9], other motivations such as saving money or benefiting one's health may also encourage individuals to shift their behaviors in a pro-environmental manner. Given these divergent definitions, there may be some behaviors that would fit into one of the provided definitions of PEB but not the other (e.g., riding a bike to work specifically to save money on gas would likely not fall under Stern's [5] definition of PEB but would be encompassed by that provided by Steg and Vlek [11]). Many recent studies [12, 18, 19, 23] have endorsed a conceptualization of PEB similar to that provided by Steg and Vlek [11], recognizing that benefits to the natural environment exist regardless of the social or psychological pathway that encourages their enactment [3, 10, 31, 32]. Expanding our understanding of PEB beyond an intention-oriented conceptualization allows for PEB to be encouraged through multiple pathways [4, 13, 31].

In conjunction with the debate on how to define PEB within the academic literature, other studies have also explored the social and psychological pathways of enacting general PEB and how these pathways influence the frequency at which different categories of behaviors are enacted [5, 23, 33, 34]. While many studies have aimed to measure general PEB through improvised unidimensional scales [17, 25, 35], such approaches fail to recognize the various factors that may inhibit or facilitate individuals' enactment of some forms of PEB in comparison to others [9, 12, 34]. As Larson et al. [23] outline, behavioral difficulty, structural factors, and social influences all may influence whether certain clusters of behaviors more or less likely to be carried out by specific individuals. For example, individuals living in a city with a well-established recycling system may easily be able to enact that specific PEB but may have difficulty enacting another behavior such as biking to work due to a lack of accessible bike lanes in the area.

As certain behaviors may be easier or more difficult to enact for various populations, it is imperative to develop tools for measuring PEB within the populations with which they will be ultimately utilized [36–39]. This is generally done by utilizing sub-scales to represent the real-world differences in how different clusters of behaviors are enacted as a result of various psychological, social, and infrastructural factors. Each sub-scale is utilized to capture a different dimension of general PEB. The heterogeneity in PEB scales developed within different populations may provide insight into this divergent dimensionality of environmentally-oriented behaviors (see the Supplementary Materials of Lange and Dewitte [19] for an extensive list of previously developed PEB scales and their various dimensions). For example, Gkargkavouzi et al. [33] found PEB to exist as six unique factors within a population of Greek citizens: civic actions, policy support, transportation choices, household setting, and consumer behavior. Alternatively, Larson et al. [23] found four dimensions of PEB within residents living in rural communities of the United States: conservation lifestyle, land stewardship, social environmentalism, and environmental citizenship. It is likely that the various social, infrastructural, and psychological contexts unique to these communities influence the dimensionality of how PEB

is perceived and enacted [9, 23, 40]. Extensive research has aimed to understand PEB across groups with varying psychological and social orientations to the natural world [10, 41, 42], but a scale measuring general PEB has not been developed across these disparate groups.

## 2.2 Scale development theory

There is an overall need for researchers to psychometrically evaluate measures prior to their use, an approach that has frequently been lacking in research examining PEB. To confidently identify what behaviors contribute to an individual's PEB, researchers must utilize scales that have demonstrated both their reliability and accuracy in measuring the construct of interest [27]. Psychometric testing, which examines a scale's reliability and validity, establishes confidence in the ability of a scale to accurately capture information on the construct researchers hope to be measuring; it is, therefore, a critical first step to conducting research. This notion of validity, or accuracy of the measure, is not absolute [43]. Although researchers often discuss the fact that the validity of a scale can be evaluated for a specific purpose, such as measuring an individual's tendency to engage in PEB, the validity of a measure may differ based on the population included in the study as well [37, 43].

Not only does one's context influence the behaviors with which they engage, constructs are understood and operationalized differently based on an individual's setting and culture [36, 37]. For this reason, researchers [27] argue for the importance of evaluating the quality of a measure in the population within which findings are to be generalized, and caution against extending research to different groups without first testing that the instrument functions equally well within that specific population [38, 39]. However, previous PEB scales have mostly been developed within either a convenience sample of university students [18] or through online survey panels [44]. When adopted for other research, these PEB scales are generally utilized in populations that vary greatly from the populations used for scale development.

## 3. Study purpose

Given the various limitations associated with measuring PEB in the field of environmental psychology, the ability to extend findings to inform psychologically-grounded interventions to promote desired behaviors is also constrained [9]. This is especially important as research interest continues to grow around processes that support the adoption of PEB generally, such as promoting behavioral spillover [45, 46]. Lange and Dewitte [19] note that self-reported PEB measures are primarily used by personality psychologists aiming to connect these measures to other psychological constructs such as environmental values and identity, emphasizing the importance of using self-reported behavioral measures that are psychologically grounded as well. Taken collectively, there is a clear need to design a PEB scale rooted in the social and psychological processes of the communities within which they will be utilized, both to promote more effective research as well as to help practitioners design more effective psychological interventions. Given this need, we aimed to develop a scale that measures a breadth of meaningful PEB while also considering the psychological dimensions of these behaviors. Specifically, we aim to establish a PEB scale that is psychometrically sound both within the general population of the United States and for a population of individuals sampled from a pro-environmental organization also based in the United States. In specifically sampling individuals from a pro-environmental organization, we surveyed individuals from the Leave No Trace Center for Outdoor Ethics, a prominent non-profit organization promoting responsible outdoor recreation behaviors in the United States [47]. This organization was intentionally chosen as the source of one of the two samples as outdoor recreation involvement and pariticpation in environmental groups has been linked to pro-environmental attitudes and behaviors in

previous research [42, 48, 49]. Given many studies aim to understand and compare PEB within populations that have various social and psychological orientation towards the natural world [25, 26, 35, 48], this study aims to develop a scale that is psychometrically stable and useful across these different populations. This study was specifically informed by the following three research questions:

RQ1: *What self-reported behaviors form a reliable and valid measure of general PEB within the unique social, cultural, and infrastructural context of the United States*?

RQ2: *What unique dimensions of general PEB are best represented as sub-scales within the general measure of PEB that is developed*?

RQ3: *What measures of general PEB form a psychometrically stable scale across a population of individuals sampled from a pro-environmental organization and a population of individuals sampled to represent the general population of the United States*?

## 4. Methods

All research procedures that involved human subjects were approved by the Pennsylvania State University Institutional Review Board (STUDY00015401). Oral or written consent was not obtained as all data were analyzed anonymously.

### 4.1 Preliminary scale development

Initial scale items were aggregated from a range of previous studies measuring various aspects of PEB [8, 17, 18, 23, 50, 51]. These studies were intentionally reviewed by experts in the field of environmental psychology, communication, and education [27] to determine the breadth of behaviors that fall within Steg and Vlek's [11] impact-oriented definition of PEB. This definition was considered simultaneously with the variety of ways previous research has attempted to characterize PEB such as activist-oriented behaviors [52], land stewardship [23, 53, 54], private-sphere behaviors [5], and socially-oriented behaviors [55]. After reviewing how PEB was measured in these previous studies, 27 initial scale items were extracted to represent a range of potential behaviors for preliminary analysis. These initial items are outlined in S1 Appendix. For these scale items, individuals were prompted to "Please rate how frequently you have participated in the following activities by selecting the appropriate point from the scale below." Seven Likert-scale response options were provided ranging from "Never" to "As frequently as possible."

These initial items were piloted within a population of 305 individuals recruited from Qualtrics market research panels [56]. All research processes for this pilot study, as well as subsequent data collection and analyses, were approved by the Pennsylvania State University Institutional Review Board. A quota sampling approach [57] was utilized to match recruited individuals to several broader demographic characteristics of the United States population as determined by the United States Census Bureau [58]. Specifically, demographic variables were matched for age, gender, and household income, each of which has been found to influence pro-environmental behavior in previous research [29].

Given that the goal of this preliminary test was to determine an initial scale for more rigorous analysis within two larger populations (one representing the general population of the United States and one sampled from a pro-environmental organization), this study phase utilized both exploratory and confirmatory techniques to eliminate redundant items as well as those that did not fit well within the psychometric properties relative to the broader scale.

**Table 1. Preliminary PEB items retained following the pilot study, including results from reliability analyses and confirmatory factor analyses.**

| Latent Construct | Item Label | Please rate how frequently you have participated in the following activities | Λ | Mean | SD |
|---|---|---|---|---|---|
| **Private Behaviors** | Priv-PEB-1 | Bought environmentally friendly and/or energy efficient products | 0.77 | 3.84 | 1.78 |
| | Priv-PEB-2 | Walked or rode a bike when traveling short distances | 0.55 | 3.38 | 2.17 |
| | Priv-PEB-3 | Reused or mended items rather than throwing them away | 0.57 | 4.19 | 1.69 |
| | Priv-PEB-4 | Composted food or yard and garden refuse | 0.66 | 2.23 | 2.24 |
| | Priv-PEB-5 | Avoided buying products with excessive packaging | 0.80 | 3.26 | 1.98 |
| | Priv-PEB-6 | Bought organic vegetables | 0.63 | 3.27 | 2.04 |
| | Priv-PEB-7 | Used rechargeable batteries | 0.54 | 3.40 | 2.07 |
| | Priv-PEB-8 | Minimized use of heating or air conditioning to limit energy use | 0.59 | 3.93 | 1.85 |
| | Priv-PEB-9 | Car-pooled when traveling to a destination | 0.61 | 2.77 | 2.15 |
| | | *Cronbach's Alpha* | 0.86 | | |
| **Public Behaviors** | Pub-PEB-1 | Talked to others in your community about environmental issues | 0.83 | 2.46 | 2.12 |
| | Pub-PEB-2 | Worked with others to address an environmental problem or issue | 0.86 | 2.11 | 2.13 |
| | Pub-PEB-3 | Participated as an active member in a local environmental group | 0.82 | 1.49 | 1.90 |
| | Pub-PEB-4 | Signed a petition about an environmental issue | 0.77 | 2.43 | 2.18 |
| | Pub-PEB-5 | Donated money to support local environmental protection | 0.85 | 2.30 | 2.17 |
| | | *Cronbach's Alpha* | 0.91 | | |

Global Fit Indices: $\Sigma^2$ = 180.34, df = 76, p<0.001; RMSEA = 0.069; SRMR = 0.048; CFI = 0.950

Specifically, descriptive statistics, exploratory factor analyses, confirmatory factor analyses, and reliability analyses were all used to examine the psychometric properties of these initial items. For more information on analysis processes and sample demographics, please see S2 Appendix. A final confirmatory factor analysis was primarily utilized at this stage to indicate preliminary discriminant validity for the scale [59]. Discriminant validity indicates the underlying sub-scale structure of a survey, which can be displayed through item loadings and latent construct relationships of a confirmatory factor analysis [60]. Items retained following this pilot study, including sub-scale structure, reliability statistics, and results from the confirmatory factor analysis are outlined in Table 1.

## 4.2 Sample populations

Following the pilot study, two primary samples were utilized to further develop the preliminary scale outlined in the previous section: a larger online sample representative of the United States population and a sample of individuals recruited from the Leave No Trace Center for Outdoor Ethics email list. For the representative sample of United States residents, 1043 individuals residing in the United States were recruited from Qualtrics market research panels [56], referred to as QUAL from here onward. Age, gender, race/ethnicity, and household income were matched using a quota sampling procedure [57] to demographic information collected from the United States census [58]. Like the other demographic variables, race/ethnicity has been found to potentially influence environmental concern in previous research [61], therefore an additional quota for race/ethnicity was intentionally added after the large percentage of White respondents was recruited from Qualtrics for the pilot study. Surveys were also distributed to an email list of 22,180 individuals provided by the Leave No Trace Center for Outdoor Ethics. This group will be referred to as LNT from here onward. Individuals were added to this list by registering as members of the organization, attending a training workshop, or taking a week-long course to become a master educator through the organization. The

online list is mostly composed of avid outdoor recreationists who participate in 8 to 12 hours of outdoor recreation per week [62].

## 4.3 Measures

Online surveys were distributed to the two populations outlined in the previous section (QUAL and LNT). These online surveys included several batteries of questions including items measuring demographic characteristics of the sample and PEB. Demographic information was collected from participants, including their gender, age, household income, level of education, and race/ethnicity. For QUAL, these demographic variables were utilized in the quota sampling procedure to ensure the representativeness of the population to the broader demographics of the United States.

Additionally, the 14 items measuring different types of PEB, outlined in Table 1, were also included in the online survey. Individuals taking the survey were prompted: "Please rate how frequently you have participated in the following activities in the past six months by selecting the appropriate point from the scale below." In contrast to the prompt utilized in the pilot study, this prompt specifically provides a time bound within which participants were asked to consider their behaviors to reduce the arbitrary nature of the measurement [63]. Six months was chosen specifically as it provided a broad enough time period for individuals to participate in PEB's that may occur less frequently (e.g., donating money to an environmental organization) while also limiting the reflection period to a reasonable period for individuals to recall. Again, seven options were provided on a Likert-scale ranging from "Never" to "As frequently as possible." These extreme end points were chosen intentionally to maximize scale variance [64].

Several additional measures were included within the online survey measuring environmental identity, outdoor recreation habits, and environmental values. These measures were utilized for an alternate study for which data were also being collected. This parallel study aimed to understand how involvement in environmentally-conscious outdoor recreation practices influenced the adoption of PEB's in other life domains. A similar population structure combining avid outdoor recreationists with a United States census-matched population was also utilized for this alternate study.

## 4.4 Data analysis

Several analytical steps were taken to further develop a psychometrically valid and reliable scale from the items initially outlined in Table 1. All data analyses were carried out in IBM SPSS 26 or R Statistical Analysis Software. Analyses that utilized confirmatory factor analyses were carried out using the AMOS extension of IBM SPSS 26.

First, an exploratory factor analysis was run for both QUAL and LNT to determine whether the two latent constructs indicated from the initial pilot sample held within the two broader samples, especially because they differed from the pilot sample regarding several demographic characteristics (outlined in the *Sample Populations* section). This analysis was carried out to specifically address research questions #1 and #2. For each of these samples, a principle component analysis (PCA) was utilized to determine the appropriate number of factors to extract from the larger scale. While not technically a form of exploratory factor analysis, this initial PCA determined the number of factors to extract by taking advantage of all available variance within the data, providing a more general approach to understanding the measures prior to determining factor loadings via further analysis. Identifying the number of factors to extract was done utilizing a combination of theoretical interpretation, examination of corresponding scree plots, and Kaiser's Rule [65]. Once the number of factors were determined, principle axis

factors (PAF) was utilized to examine how individual items related to the underlying latent constructs represented by the scale. This sequence of steps (PCA followed by PAF) followed the exploratory factor analysis approach outlined by [66]. A cutoff value for factor loadings of 0.32 was utilized, as items below are less likely to have a statistically meaningful relationship with the associated latent construct [67].

Second, in order to further address research questions #1 and #2, two confirmatory factor analyses were run, one for QUAL one for LNT, to establish the factors structure of the PEB items. Like the confirmatory factor analysis conducted within the pilot study, the $\Sigma^2$ statistic was used to assess model fit [68]. Additionally, other global fit indices utilized to assess model fit were: RMSEA $\leq 0.10$ [68]; SRMR $\leq 0.08$ [68]; and CFI $\geq 0.90$ [69]. Also in parallel to the pilot study, factor loading were considered adequate if statistically significant and with loadings over 0.30 [70]. Bias-corrected confidence intervals (95% confidence interval computed by 5000 bootstrap resamples) helped to minimize the likelihood of Type 1 Error for factor loadings [71].

Third, to explore the psychometric stability of the final measure between a general population of individuals living in the United States and members of the Leave No Trace Center for Outdoor Ethics (addressing research question #3), invariance testing was utilized to compare the psychometric equivalence of the measure between the QUAL and LNT groups. On the importance of exploring measurement invariance between groups for which the measure will ultimately be utilized, [28] state, "meaning is essentially conventionalized, and so different groups can apply different meanings to the same cognition or behavior. Appropriate and proper comparison of a construct between groups or across times, therefore, depends first on ensuring equivalence of meaning of the construct" (p. 72). Therefore, configural, metric, and scalar invariance [28] were explored between the LNT and QUAL groups. This is done through an iterative process of sequentially imposing greater constraints on a multi-group CFA that incorporates both groups of interest [72]. Configural invariance explored whether the basic item-construct structure was the same between the two groups. Broadly, configural invariance suggests the same items load on to the same latent constructs across the two groups of interest [28, 73]. This was determined by assessing model fit using the same fit indices outlined for the confirmatory factor analysis in the previous paragraph. Metric invariance then constrained item loadings to be equivalent between the two groups and determined whether this constraint significantly reduced model fit when compared to the configural model. Theoretically, metric invariance suggests that not only the same items load on to the same latent constructs across the two groups (as explored in configural invariance), but the same items contribute to the same latent construct in a similar pattern across the two groups [28, 73]. Finally, scalar invariance constrained item intercepts between QUAL and LNT. Scalar invariance explores whether differences in latent constructs between the two groups adequately captures mean differences in the shared variance across the measured items (i.e., higher levels of a measured item in one group result in a higher level of the corresponding latent construct) [28, 73]. Reduction in model fit for the scalar invariance model was compared to the metric invariance model, with all outlined procedures matching those outlined in Putnick and Bornstein [28].

Since $\Sigma^2$ is sensitive to large sample sizes, reduction in model fit was assessed utilizing alternative fit indices, specifically the cutoff measures developed by Cheung and Rensvold [74] and Chen [75]. Changes in fit indices should not exceed -0.010 for CFI and 0.015 for RMSEA. Additionally, for SRMR, changes should be less than or equal to 0.030 for metric invariance and 0.015 for scalar invariance. If model fit failed to meet thresholds of advancing to the next strictest model, constraints were selectively released based off meaningful differences between the two groups. Model fit was then reassessed for the partially restricted model.

**Table 2. Demographic characteristics of the two collected samples.**

| Demographic Variables | | n | Percentage of Sample | n | Percentage of Sample |
|---|---|---|---|---|---|
| | | LNT Mean age = 47.3 (SD = 16.2) | | QUAL Mean age = 45.6 (SD = 17.1) | |
| Gender | Female | 847 | 42.8 | 541 | 51.9 |
| | Male | 1019 | 51.5 | 498 | 47.7 |
| | Non-binary | 21 | 1.1 | 4 | 0.4 |
| | Missing | 91 | 4.6 | 0 | 0.0 |
| Ethnicity | White | 1725 | 87.2 | 643 | 61.6 |
| | Hispanic or Latino/Latina/Latinx | 47 | 2.4 | 188 | 18.8 |
| | Black or African American | 8 | 0.4 | 127 | 12.2 |
| | Native American, American Indian, or Alaska Native | 16 | 0.8 | 10 | 1.0 |
| | Asian or Pacific Islander | 38 | 1.9 | 58 | 5.6 |
| | Other | 40 | 2.0 | 17 | 1.6 |
| | Missing | 104 | 5.3 | 0 | 0.0 |
| Household Income | Less than $10,000 | 52 | 2.6 | 76 | 7.3 |
| | $10,000-$19,999 | 73 | 3.7 | 74 | 7.1 |
| | $20,000-$29,999 | 97 | 4.9 | 107 | 10.3 |
| | $30,000-$39,999 | 133 | 6.7 | 85 | 8.1 |
| | $40,000-$49,999 | 115 | 5.8 | 82 | 7.9 |
| | $50,000-$59,999 | 125 | 6.3 | 84 | 8.1 |
| | $60,000-$69,999 | 147 | 7.4 | 82 | 7.9 |
| | $70,000-$79,999 | 119 | 6.0 | 79 | 7.6 |
| | $80,000-$89,999 | 114 | 5.8 | 51 | 4.9 |
| | $90,000-$99,999 | 112 | 5.7 | 47 | 4.5 |
| | $100,000-$149,999 | 390 | 19.7 | 176 | 16.9 |
| | More than $150,000 | 312 | 15.8 | 100 | 9.6 |
| | Missing | 189 | 9.6 | 0 | 0.0 |
| Education | Elementary | 0 | 0 | 1 | 0.1 |
| | Some high school | 1 | 0.1 | 33 | 3.2 |
| | GED or high school graduate | 54 | 2.7 | 213 | 20.4 |
| | Some college or technical school | 364 | 18.4 | 373 | 35.8 |
| | Four-year college graduate | 785 | 39.7 | 262 | 25.1 |
| | Graduate degree | 681 | 34.4 | 161 | 15.4 |
| | Missing | 93 | 4.7 | 0 | 0.0 |

## 5. Results

### 5.1 Sample characteristics and descriptive statistics

Of the distributed surveys, 1043 completed surveys were returned for QUAL while 1978 surveys were returned for LNT. QUAL was primarily white (61.6%), had a slight majority of females (51.9%), and had a mean age of 45.6 years old. Like QUAL, LNT was primarily white (though at a much higher percentage at 87.2%) and had a similar mean age of 47.3 years old. LNT deviated from QUAL in being primarily male (51.5%). Additionally, LNT generally had a higher household income and education level than those in QUAL. Detailed demographic information for both samples is provided in Table 2.

Descriptive statistics for the 14 items measuring PEB are outlined in Table 3, and item means are outlined for both QUAL and LNT. As expected, scores on the PEB measures were generally

**Table 3. Descriptive statistics for QUAL and LNT for PEB items.**

| Item Label | QUAL | | | LNT | | |
|---|---|---|---|---|---|---|
| | Mean | SD | % Missing | Mean | SD | % Missing |
| Priv-PEB-1 | 3.85 | 1.64 | 0 | 4.94 | 1.21 | 10.5 |
| Priv-PEB-2 | 3.54 | 2.00 | 0 | 4.46 | 1.63 | 10.4 |
| Priv-PEB-3 | 4.10 | 1.68 | 0 | 4.99 | 1.20 | 10.4 |
| Priv-PEB-4 | 2.72 | 2.18 | 0 | 3.72 | 2.31 | 10.7 |
| Priv-PEB-5 | 3.46 | 1.86 | 0 | 4.36 | 1.55 | 10.4 |
| Priv-PEB-6 | 3.42 | 1.92 | 0 | 4.06 | 1.78 | 10.4 |
| Priv-PEB-7 | 3.42 | 1.97 | 0 | 3.56 | 1.92 | 10.6 |
| Priv-PEB-8 | 3.85 | 1.74 | 0 | 4.62 | 1.44 | 10.5 |
| Priv-PEB-9 | 2.90 | 2.06 | 0 | 3.82 | 1.84 | 10.8 |
| Pub-PEB-1 | 2.56 | 1.99 | 0 | 4.04 | 1.67 | 10.7 |
| Pub-PEB-2 | 2.37 | 1.94 | 0 | 3.54 | 1.85 | 10.8 |
| Pub-PEB-3 | 1.94 | 2.00 | 0 | 2.98 | 2.14 | 10.5 |
| Pub-PEB-4 | 2.51 | 2.14 | 0 | 3.31 | 2.12 | 10.8 |
| Pub-PEB-5 | 2.43 | 1.99 | 0 | 3.58 | 1.88 | 10.6 |

higher in LNT, further supporting prior work showing that individuals invested in environmental organizations and outdoor recreation self-report higher levels of PEB [25, 35]. While there was no missing data in QUAL, missing data was reported across the PEB items for LNT. Little's MCAR Test [76] was utilized to explore whether data for the PEB items was missing randomly within LNT, an assumption that was confirmed by the analysis ($\Sigma^2 = 270.85$, df = 269, p = 0.46). Once data were confirmed to be missing completely at random for LNT, the individuals with missing data on PEB measures were deleted listwise, leaving 1719 individuals for analysis.

## 5.2 Exploratory factor analyses

Exploratory factor analyses were utilized to further understand the underlying psychological dimensions of the 14 PEB items retained from the pilot sample in both the QUAL and LNT samples. Utilizing the Kaiser Rule [65] and examining the corresponding scree plot for each of the two PCAs, both samples suggested that two underlying latent constructs existed within the broader scale measuring PEB. For LNT, eigenvalues were 4.63 and 1.41 for the first two factors, respectively, with all other factors loading below the cutoff value of 1. For QUAL, the eigenvalues were 6.42 and 1.43. Similarly, all other factors failed to meet the threshold of exceeding 1. This statistical evidence also aligned with previous empirical evidence [24, 77] and with our pilot study that indicated two latent constructs existed within the concept of PEB: Private Behaviors and Public Behaviors. Eigenvalues for all extracted factors for both PCAs are outlined in Table 4.

Once the number of factors to extract were determined using PCA, PAF was utilized to further examine the nature of these latent constructs and how individual items loaded onto them for each population. An oblique rotation was utilized as the two latent factors were highly correlated with each other (0.61 for QUAL and 0.55 for LNT).

**Table 4. Eigenvalues for each factor extracted from LNT sample and GEN sample.**

| LNT | 4.63 | 1.41 | 0.99 | 0.91 | 0.85 | 0.79 | 0.72 | 0.69 | 0.64 | 0.57 | 0.53 | 0.48 | 0.43 | 0.35 |
|---|---|---|---|---|---|---|---|---|---|---|---|---|---|---|
| QUAL | 6.42 | 1.43 | 0.79 | 0.69 | 0.67 | 0.63 | 0.62 | 0.53 | 0.49 | 0.42 | 0.40 | 0.35 | 0.30 | 0.26 |

**Table 5. Item loadings on two extracted factors for both QUAL and LNT samples.**

|  | QUAL | | LNT | |
|---|---|---|---|---|
|  | Factor 1 | Factor 2 | Factor 1 | Factor 2 |
| **Priv-PEB-1** | 0.74 | 0.04 | 0.67 | 0.01 |
| **Priv-PEB-2** | 0.41 | 0.22 | 0.45 | 0.05 |
| **Priv-PEB-3** | 0.72 | -0.09 | 0.56 | -0.04 |
| **Priv-PEB-4** | 0.29 | 0.36 | 0.32 | 0.07 |
| **Priv-PEB-5** | 0.68 | 0.09 | 0.70 | 0.00 |
| **Priv-PEB-6** | 0.42 | 0.25 | 0.51 | 0.02 |
| **Priv-PEB-7** | 0.43 | 0.17 | 0.28 | 0.15 |
| **Priv-PEB-8** | 0.67 | -0.08 | 0.56 | -0.06 |
| **Priv-PEB-9** | 0.29 | 0.37 | 0.37 | 0.10 |
| **Pub-PEB-1** | 0.08 | 0.77 | 0.16 | 0.59 |
| **Pub-PEB-2** | 0.01 | 0.84 | 0.02 | 0.77 |
| **Pub-PEB-3** | -0.12 | 0.92 | -0.10 | 0.83 |
| **Pub-PEB-4** | 0.10 | 0.67 | 0.31 | 0.37 |
| **Pub-PEB-5** | 0.06 | 0.75 | 0.17 | 0.48 |

Item factor loadings for each latent construct are outlined in Table 5. After examining the item loadings on each latent factor, three problematic items were identified between the QUAL and LNT samples. For QUAL, Priv-PEB-4 and Priv-PEB-9 both cross-loaded considerably, with a slightly higher loading on the factor that seemed to align with Public Behaviors rather than Private Behaviors. This contrasted with what was indicated by the findings from the pilot study. Specifically, Priv-PEB-4 asked about composting behavior, a behavior that is constrained by infrastructural availability for some individuals (e.g., those living in urban environments), which could pose a significant limitation to performing a behavior even if behavioral intent was present [31]. Additionally, Priv-PEB-9 inquired about carpooling behaviors. While in some regards this behavior could take place in a private setting, it also required the participation of others. Given this practical justification and inconsistent statistical performance across the two groups, both items were dropped from further analysis. Additionally, Priv-PEB-7, which asked about the use of rechargeable batteries, did not meet the pre-determined loading threshold of 0.32 [67] for the LNT sample. Given the goal of this study is to develop a scale measuring PEB that can be generalized between general and pro-environmental groups within the United States, this item was also dropped.

In broadly examining the remaining 11 item loadings, they aligned with the two latent factors of Private Behaviors and Public Behaviors observed in the pilot study. With this, the six remaining items from the Private Behaviors sub-scale and the five items making up the Public Behaviors sub-scale established from the pilot study were maintained for further analysis.

## 5.3 Confirmatory factor analyses

Confirmatory factor analyses were utilized to analyze the relationship between the 11 items retained after the exploratory factor analysis for both the QUAL and LNT populations. Model fit indices and factor loadings are outlined for each population in the following paragraphs. Labels for the two latent constructs, Private Behaviors and Public Behaviors, were retained from the pilot study for the confirmatory factor analysis. Each of these corresponding labels continued to capture the theoretical concept represented by their corresponding items effectively.

For QUAL, $\Sigma^2$ indicated poor model fit ($\Sigma^2$ = 211.082, df = 43, p<0.001), a result likely linked to the large sample size. Other global fit indices alternatively indicated good model fit though: RMSEA = 0.061; SRMR = 0.044; CFI = 0.969. All factor loadings also met appropriate thresholds (0.30) on the corresponding latent construct and were significant. For LNT, $\Sigma^2$ values also indicated poor model fit ($\Sigma^2$ = 415.713, df = 43, p<0.001), though the large LNT sample size also likely contributed to this result [68]. Alternative global fit indices also met necessary thresholds: RMSEA = 0.071; SRMR = 0.047; CFI = 0.929. Like QUAL, all items loaded on the related latent constructs at appropriate levels (0.30) while also being statistically significant. With this, both original confirmatory factor analysis models were retained without modification for QUAL and LNT. Models for both QUAL and LNT populations are outlined in Figs 1 and 2, respectively.

## 5.4 Measurement invariance

Invariance testing was conducted between QUAL and LNT to explore the psychometric stability of the PEB measures and their associated latent constructs across the two populations. The configural invariance model was found to have appropriate model fit when examining all fit indices aside from the $\Sigma^2$ measure ($\Sigma^2$ = 626.786, df = 86, p<0.001; RMSEA = 0.048; SRMR = 0.0471; CFI = 0.950). Given these appropriate fit statistics, configural invariance for the PEB measures between the two groups was supported. In comparing the metric invariance model to the configural invariance model, corresponding changes in model fit fell within the pre-established thresholds for RMSEA and SRMR measures but exceeded the appropriate level of change for the CFI measures. The fit indices for the metric invariance model were $\Sigma^2$ = 800.686, df = 97, p<0.001; RMSEA = 0.051; SRMR = 0.0687; CFI = 0.935. The -0.015 change in CFI exceeded the threshold of -0.010 established by both Cheung and Rensvold [74] and Chen [75].

Given the PEB measures and the associated latent constructs were found to be metric non-invariant, single item loadings were released in a stepwise manner to understand whether partial metric invariance could be established. These items were released specifically by examining potential theoretical differences between the QUAL and LNT populations. Ultimately, constraints on four item loadings were released: Priv-PEB-2, Pub-PEB-1, Pub-PEB-3, and Pub-PEB-5. Priv-PEB-2 specifically addresses walking and biking behaviors in commuting to nearby destinations. Since prior research has indicated that the LNT population regularly partakes in high levels of outdoor recreation on a weekly basis [62], the unique nature of the LNT population may have resulted in loading differences for this item. Additionally, Pub-PEB-1 and Pub-PEB-3 may be directly influenced by the nature of individuals' involvement with the Leave No Trace Center for Outdoor Ethics. These items specifically address whether individuals talk to others about environmental issues and whether they participate as a regular member in an environmental organization. Given many individuals within the LNT sample are involved with the Leave No Trace Center for Outdoor Ethics as educators, this may have resulted in unique difference for factor loadings on these two items as well when compared to a general population. Lastly, LNT had much higher household income levels than QUAL, potentially influencing difference in how individuals rated frequency of participation in Priv-PEB-5 which asks about donating money to support environmental protection.

When item loadings were released between groups for these four items, partial metric invariance was supported as changes in fit indices fell within all appropriate thresholds ($\Sigma^2$ = 742.186, df = 93, p<0.001; RMSEA = 0.050; SRMR = 0.0615; CFI = 0.940). Releasing factor loadings for four items was considered appropriate as previous research has indicated that constraints should be maintained on at least half of the items for partial metric invariance to

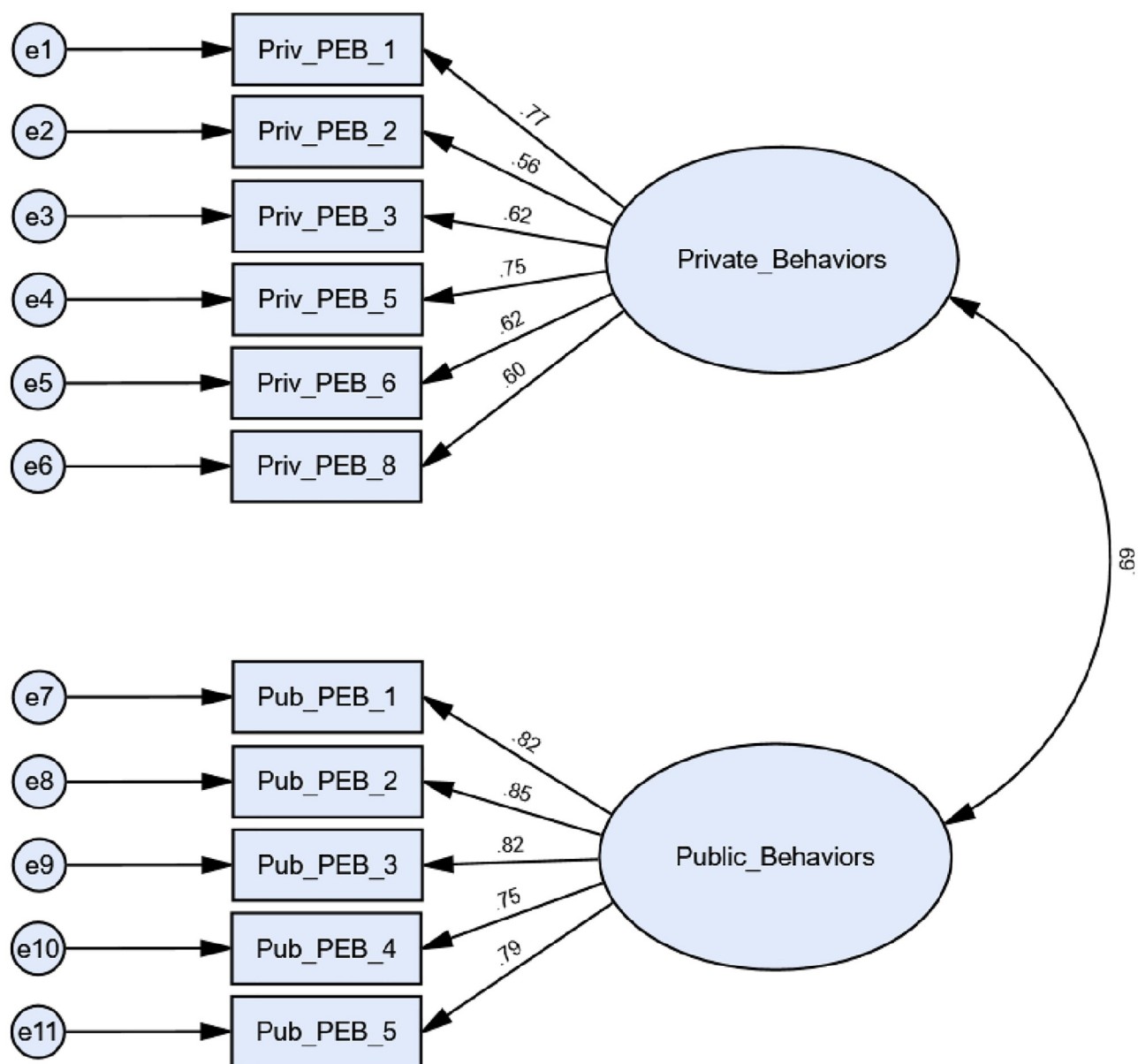

**Fig 1. Confirmatory factor analysis diagram for QUAL with standardized estimates; global fit indices: $\Sigma^2$ = 211.082, df = 43, p<0.001; RMSEA = 0.061; SRMR = 0.044; CFI = 0.969.**

be confirmed [78, 79]. Partial metric invariance indicates that item loadings were generally comparable across the LNT and QUAL groups, excluding the four items that were allowed to vary between groups (theoretical justification in why these items may not load in a similar pattern across groups being outlined previously). Given poor model fit and change in fit indices greatly exceeding appropriate thresholds when constraining for item intercepts, it was confirmed that the items were scalar noninvariant across the two groups. Model fit indices are further outlined in Table 6. Achieving partial metric invariance, but not scalar invariance, indicates that the proposed scale behaves similarly across the two groups, but some limitations exist in this similarity. The four indicated items (Priv-PEB-2, Pub-PEB-1, Pub-PEB-3, and

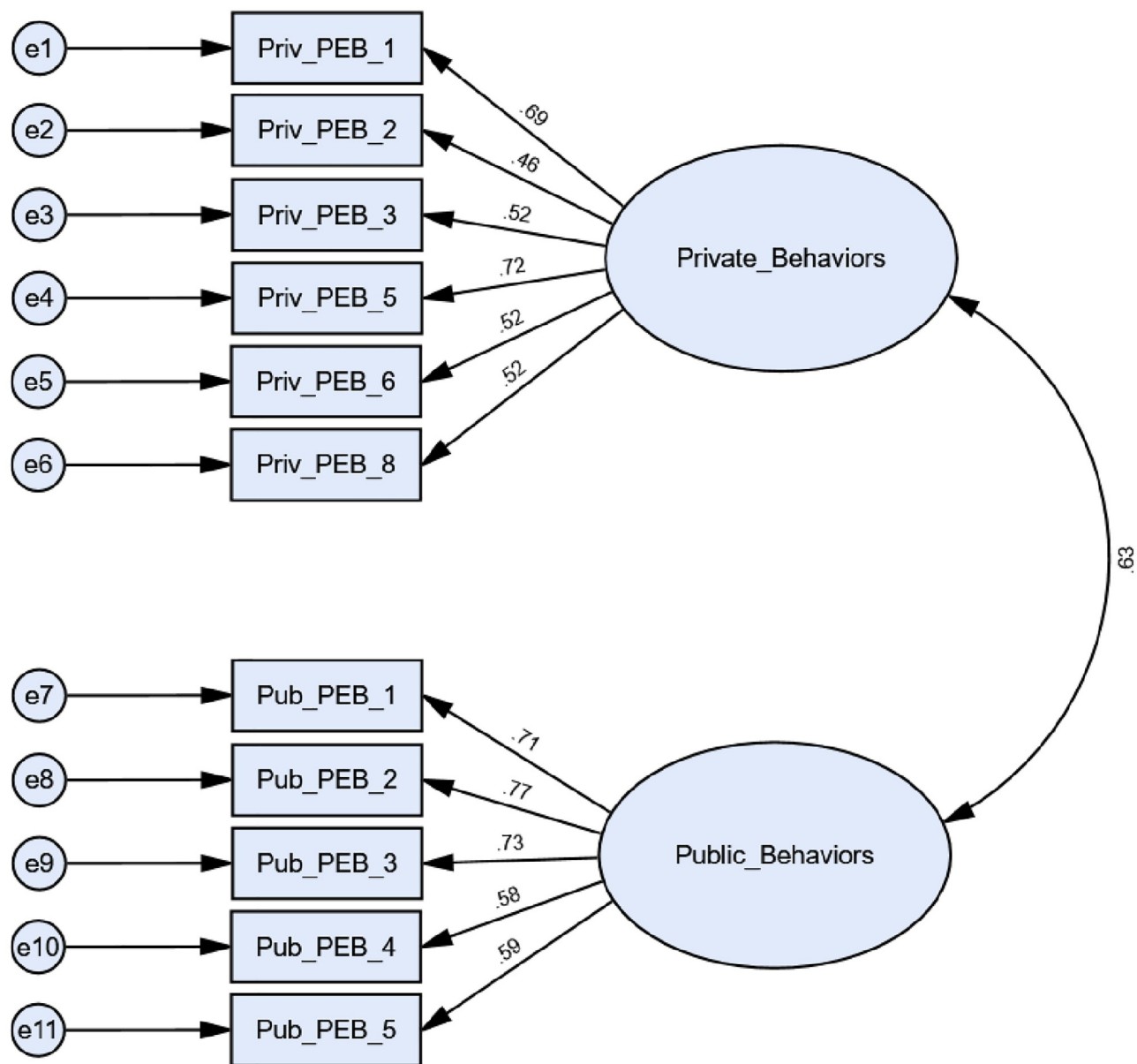

**Fig 2. Confirmatory factor analysis diagram for LNT with standardized estimates; global fit indices:** $\Sigma^2$ = 415.713, df = 43, p<0.001; RMSEA = 0.071; SRMR = 0.047; CFI = 0.929.

Pub-PEB-5) load differently on the Private Behaviors and Public Behaviors latent constructs. Additionally, failing the test for scalar invariance indicates that an increase or decrease in an item's intercept does not uniformly influence that latent construct between the LNT and QUAL groups. While partial metric invariance indicates that the developed scale behaves similarly across the two groups of interest, limitations to this similarity should be acknowledged.

## 5.5 Reliability analysis

Scale reliabilities were calculated using Cronbach's Alpha [80] for each of the Private Behaviors and Public Behaviors sub-scales for both QUAL and LNT. The reliability measure for the

**Table 6. Models testing measurement invariance for QUAL and LNT samples.**

| Model | $\Sigma^2$ | df | p | RMSEA | SRMR | CFI |
|---|---|---|---|---|---|---|
| Configural | 626.786 | 86 | <0.001 | 0.048 | 0.0471 | 0.950 |
| Partial Metric* | 742.186 | 93 | <0.001 | 0.050 | 0.0615 | 0.940 |
| Metric | 800.686 | 97 | <0.001 | 0.051 | 0.0687 | 0.935 |
| Scalar | 1433.003 | 108 | <0.001 | 0.067 | 0.1026 | 0.877 |

*Partial metric invariance released constraints on four item loadings: Priv-PEB-2, Pub-PEB-1, Pub-PEB-3, and Pub-PEB-5

Private Behaviors sub-scale in QUAL was 0.82, while the reliability measure for the Public Behaviors sub-scale in the same group was 0.90. For LNT, the Private Behaviors sub-scale had a reliability of 0.74. The Public Behaviors sub-scale had a reliability of 0.80 in LNT.

## 6. Discussion

This study provides researchers and practitioners with a reliable and valid scale to measure general PEB in the United States across groups with different levels of involvement in pro-environmental organizations or activities. Results mostly supported scale reliability and validity for both the LNT and QUAL groups. Additionally, partial metric invariance also indicated a level of consistency in how PEB was perceived across the different populations as well. These findings may allow for scale comparability across these different groups, though it should be noted that not all factor loadings and intercepts related to latent constructs in the same manner across groups. This scale overcomes limitations generally attributed to other measures of PEB in prior research, primarily a lack of rigorous psychometric testing in populations of interest to encourage use across studies. In this generalized scale, the sub-scale structure is split between private and public behaviors. This represents a simplification in how behaviors have been categorized when compared to previous scales measuring PEB, which have included a larger number of behavioral sub-scales [18, 23, 33, 81].

Indicators of reliability and validity for the developed scale were appropriate across both a population of individuals involved in a pro-environmental organization and a general population of individuals in the United States. Establishing these metrics between the two groups represents an important step forward in developing a scale measuring self-reported PEB. This is especially important as many studies measure behaviors within or between populations that have a range of involvement in environmental organizations, outdoor recreation activities, or psychosocial factors related to the behaviors themselves [25, 26, 44]. The establishment of partial metric invariance for the developed scale across these two populations further indicates that there is a level of psychometric stability across these divergent groups as well [28]. It is helpful to establish this consistency in the psychometric performance of the scale as differences in self-reported PEB across groups in future studies can more definitively be linked to changes in variables of interest rather than unaccounted for differences in how the scale is interpreted between groups [28]. Specifically, for this study, various practical differences between the QUAL and LNT groups may explain why full metric invariance was not achieved.

In developing this novel measure, a bi-dimensional sub-scale structure, in line with the public/private split highlighted by some other PEB research [24, 77], is supported as a psychometrically sound way to understand the multi-dimensional nature of PEB across these different groups. This is somewhat unsurprising given the significant role social norms have been found to play in influencing how PEB is enacted in various circumstances [31, 82, 83]. Public behaviors are generally social in nature, and this may be a contributing factor to the underlying

sub-scale structure. Despite the influence social norms have been found to have on PEB, they have not been the primary factor determining the structure of a widely used PEB scale previously. Though this sub-scale structure has not been represented in previous developed scales (in contrast to more theoretical work on the nature of PEB; [31, 82, 83]), measuring general PEB through these two dimensions was determined to be psychometrically consistent across LNT and QUAL groups.

In looking to other PEB measures utilizing sub-scales that have received widespread use previously, the bi-dimensional structure of the scale developed in this study generally represents a simplification in categorizing self-reported behaviors. Markle [18] asserts PEB should be measured via four sub-scales (conservation, environmental citizenship, food, and transportation), while Kaiser [81] asserts there should be seven sub-scales (prosocial behavior, ecological garbage removal, water and power conservation, ecologically aware consumer behavior, garbage inhibition, volunteering in nature protection activities, and ecological automobile use). Evidence from this study suggests that these previously established measures may over specify the multi-dimensional nature of PEB when understanding it in the broad context of the general United States population. For example, in an alternative conceptualization of PEB by Larson et al. [23], "land stewardship" was an important aspect of PEB that is notably absent from the scale developed in this study. Given that Larson et al. [23] developed their scale within a rural population in the United States, accessibility or ownership of natural areas is likely more widespread throughout their population. When understanding PEB within the broader population of the United States, these behaviors may not be as commonplace due to restricted access to natural areas. For example, composting behavior was eventually removed from our scale due to its cross-loading between sub-scales amongst the QUAL group in the exploratory factor analysis, indicating there may be some confounding factors influencing how this behavior is perceived and enacted. This aligns with findings by Huddart-Kennedy et al. [51], suggesting that land stewardship behaviors (a broader PEB category that can include composting) are enacted less frequently in populations outside of rural areas. While generalizing to the United States population may lead to a reduced level of nuance within the sub-scale structure of the developed measure, two dimensions were found to be reliable and valid across both the LNT and QUAL populations. In comparison to previously developed scales, the scale developed here may oversimplify the nature of PEB to a degree. There is certainly less breadth of behaviors represented in the final scale in comparison to the initial 27 items utilized in the pilot study (outlined in S1 Appendix). This simplification is somewhat of a trade-off for other aspects of the developed scale that may be beneficial: the generally stable psychometric properties across groups with varying environmental orientations and its brief nature. Additionally, findings from this study further support the assertion made by Larson et al. [23] that PEB is not a unidimensional construct and therefore should not be measured as such. This simplified practice diminishes the behavioral and psychological complexity of PEB, a layer of intricacy that should not be lost if future studies are to most effectively understand PEB.

## 7. Limitations

Several limitations must be acknowledged when drawing conclusions and implications from the findings of this study. Firstly, the PEB scale was developed solely within populations residing in the United States. The behaviors and associated sub-scale structure are therefore representative of how PEB is perceived and enacted in the United States. Application of this scale to other populations should be done with care and additional psychometric testing, acknowledging that the associated social and psychological factors influencing PEB may differ across cultures and locations. This newly developed scale may provide direction in how to measure PEB

in other populations, but measurement invariance should be established first [28]. Establishing such consistency across different countries and regions represents a potential step forward for future research.

Secondly, while partial metric invariance was established between the LNT and QUAL groups, these findings indicate that divergences remain in how the developed scale is perceived and interpreted between the two different groups. While this degree of psychometric stability between groups presents a considerable step forward in developing a scale that can be used across populations with differing levels of involvement in environmentally-oriented causes, there remain limitations to the scale's performance as well. Item loadings on latent constructs across groups were not fully stable, and difference in intercepts did not correspond to uniform differences in latent constructs across the two groups. This indicates that future applications of this measure across groups with varying involvement in pro-environmental organizations may need to consider that some differences may be a result of how sub-scales are perceived rather than true differences in behavior.

Thirdly, it should be noted that this study did not utilize a new sample to further confirm scale structure after the primary analyses done across the QUAL and LNT groups. Utilizing an additional sample to determine the scale structure would further support the psychometric stability of the developed PEB scale. Despite the potential benefits of recruiting an additional population of individuals to test scale properties, the three populations (pilot study group, QUAL, and LNT) utilized throughout the presented study suggest initial evidence for utilizing the items and sub-scale structure in future research. While the presented scale could further be confirmed in a new group of individuals, this step may only add marginal evidence for scale stability and structure. Given limited time and resources allocated to the scale development process, this additional step may not be justified by the amount of additional information that it would provide.

## 8. Conclusion

While a range of studies have been focused on how to measure self-reported PEB previously, this study represents a step forward in developing a psychometrically sound scale that can be more readily utilized in populations with a range of orientations towards the natural environment. While some previous PEB studies have aimed to establish measurement consistency across different populations such as between individuals residing in different countries [8], this scale development process took the novel approach of developing a scale that is explicitly designed to be consistent across populations with different levels of involvement with a pro-environmental organization. Additionally, the range of behaviors measured relative to the scale's short length (11 items) represents a tool for future researchers to readily incorporate into future studies without significantly increasing measurement burden.

Considering the plethora of research focused on understanding and encouraging PEB within the field of environmental psychology, the use of a consistent scale across these studies can aid greatly in more holistically understanding trends across and between studies. This, in turn, allows for researchers and practitioners to better develop a body of collective knowledge on how to encourage PEB in various populations. Developing consistent, reliable, and valid measurement practices represents a necessary step towards achieving that goal.

## Supporting information

**S1 Data. Anonymized data utilized in study procedures.**
(CSV)

**S1 Appendix.**
(DOCX)

**S2 Appendix.**
(DOCX)

**S3 Appendix.**
(DOCX)

## Acknowledgments

The views expressed in this article are the responsibility of the authors and do not necessarily represent the opinions or policy of the National Park Service.

## Author Contributions

**Conceptualization:** Timothy J. Mateer, Zachary D. Miller, Ben Lawhon, Jennifer P. Agans, B. Derrick Taff.

**Data curation:** Timothy J. Mateer, Jennifer P. Agans, B. Derrick Taff.

**Formal analysis:** Timothy J. Mateer, Theresa N. Melton, Zachary D. Miller, Jennifer P. Agans.

**Funding acquisition:** Ben Lawhon, B. Derrick Taff.

**Investigation:** Zachary D. Miller, Ben Lawhon.

**Methodology:** Timothy J. Mateer, Theresa N. Melton, Zachary D. Miller, Ben Lawhon, Jennifer P. Agans.

**Project administration:** Timothy J. Mateer, Ben Lawhon, B. Derrick Taff.

**Supervision:** Ben Lawhon, B. Derrick Taff.

**Writing – original draft:** Timothy J. Mateer, Theresa N. Melton.

**Writing – review & editing:** Timothy J. Mateer, Theresa N. Melton, Jennifer P. Agans, B. Derrick Taff.

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
