## [Decision Letter · Decision Letter 0]

11 May 2022

PONE-D-21-27321A multi-dimensional measure of pro-environmental behavior for use across populations with varying levels of environmental involvement in the United StatesPLOS ONE

Dear Dr. Mateer,

Thank you for submitting your manuscript to PLOS ONE. After careful consideration, we feel that it has merit but does not fully meet PLOS ONE’s publication criteria as it currently stands. Therefore, we invite you to submit a revised version of the manuscript that addresses the points raised during the review process. The two reviewers agree on the importance and need to have better tools to measure PEBs. I concur.  Your are making a useful contribution to that. Please address all of their comments, but I would highlight a couple of points. 1. Please send a much more concrete manuscript. 2. I think that apart from the scale, by itself, this paper brings to the table a key issue in the measurement of PEBs. However, It seems to me you may be overestimating the pervasiveness of your scale. I need you to improve the discussion of how and when this scales is better than other approaches and acknowledge very clearly its limitations, most notably, vis a vis, behavioral measures.I look forward to receiving the the revised manuscript.

We look forward to receiving your revised manuscript.

Kind regards,

Carlos Andres Trujillo, PhD

Academic Editor

PLOS ONE

**Journal requirements:**

3. Please include your tables as part of your main manuscript and remove the individual files. Please note that supplementary tables (should remain/ be uploaded) as separate ""supporting information"" files.

Reviewers' comments:

Reviewer's Responses to Questions

**Comments to the Author**

1. Is the manuscript technically sound, and do the data support the conclusions?

Reviewer #1: Partly

Reviewer #2: Partly

2. Has the statistical analysis been performed appropriately and rigorously? 

Reviewer #1: Yes

Reviewer #2: Yes

3. Have the authors made all data underlying the findings in their manuscript fully available?

Reviewer #1: No

Reviewer #2: Yes

4. Is the manuscript presented in an intelligible fashion and written in standard English?

Reviewer #1: No

Reviewer #2: Yes

5. Review Comments to the Author

Reviewer #1: Dear authors, I have the following comments to your submission. I hope you find these comments helpful to improve your manuscript and advance in the process.

1. Is the manuscript technically sound, and do the data support the conclusions?

Although the authors present several analyses and psychometric testing, to examine the scale’s reliability and validity, there exists some information to clarify:

-On page 18 you say: "a final confirmatory analysis was utilized at this stage (pilot) to indicate preliminary discriminant validity for the scale" (Brown, 2015). However, this is unclear because you do not explain how this analysis indicates discriminant validity (page 26). Moreover, the analysis is not presented in the results. So, the reader does not know the discriminant validity of the scale.

About the configural, metric, and scalar invariance:

-I recommend extending the explanation of the invariance metrics to understand better how they work.

-Some literature about invariance metrics refers to the need to run a Multigroup CFA even if the model fits well in each group. I think you should explain in detail if this analysis was done or not and why.

-On page 28 it says:

"When item loadings were released between groups for these four items, partial metric invariance was supported as changes in fit indices fell within all appropriate thresholds (ꭓ2=742.186, df=93, p<0.001; RMSEA=0.050; SRMR=0.0615; CFI=0.940). Releasing factor loadings for four items was considered appropriate as previous research has indicated that constraints should be maintained on at least half of the items for partial metric invariance to be confirmed (Steenkamp & Baumgartner, 1998; Vandenberg & Lance, 2000)".

I would suggest that at any point, before or after this paragraph, could be useful to know what partial invariance means, why it is a valid metric, and when to use it.

-Although the analysis of scalar invariance failed, you do not present the implications of this result. In the discussion, it says the scale is reliable and valid, however, this is vague until you clarify the results of scalar invariance and its implications in the whole analysis of validity.

2. Has the statistical analysis been performed appropriately and rigorously?

Yes, however, I wonder whether other analyses to test the validity of a scale could be useful to robust your paper, otherwise, could you explain why those analyses do not fit with the purpose of your study and, therefore, they were omitted? For example, discriminant validity and criterion-related validity (Kaiser & Biel, 2000; Kaiser, 1998).

3. Have the authors made all data underlying the findings in their manuscript fully available?

-Even though the purpose of the paper is "to develop a scale that measures a breadth of meaningful PEB, considering the psychological dimensions of these behaviors", the scale indeed does not include a breadth of PEB (e.g. water conservation and recycling behaviors are omitted). I suggest reporting the initial 27 items you use at the starting point to do the pilot study. I understand these items come from the previous literature about PEB. So, it could be useful to know the range of initial behaviors to understand what were the initial items dropped, and why some pro-environmental domains are missed in the whole analysis. I think it is confusing to present a scale about a breadth of meaningful PEB with important domains of private behaviors out. In the discussion, you say that your study suggests that previously established measures may over-specify the multidimensional nature of PEB. Probably, your scale may do an oversimplification because some domains are missing.

4. Is the manuscript presented in an intelligible fashion and written in standard English?

-There are some errors in the writing (duplicated words, typos, and wrong written words; e.g. pages 16 and 27)

Other particular comments:

-Page20: When you say: "Additionally, the 14 items measuring different aspects of PEB, outlined in Table 1", it is confusing the writing, and it seems imprecise to say different aspects. I suggest saying different PEB because you are measuring PEB, not aspects of them.

-I would like to know why the 6 months is a proper time-bound to consider the behaviors people performed? Why not 1 month, 3 months?

-Page 24: "For LNT, eigenvalues were 4.61 and 1.42 for the first two factors". -> These are not the same values that appear in Table 4. Please, match them.

-Page 24: "An oblique rotation was utilized as the two latent factors were highly correlated with each other (0.61 for GEN and 0.55 for LNT)". -> Please, keep the same letters to identify the populations (GEN vs QUAL) because it could be confusing for the reader.

-In the Discussion, you say: "This study provides researchers and practitioners with a reliable and valid scale to measure general PEB in the United States across groups with different levels of involvement in

pro-environmental organizations or activities. Results suggested strong evidence for scale reliability and validity for both the LNT and QUAL groups". -> this affirmation seems a bit strong taking into account the limitations of the study.

-In the Discussion (page 30) the sentence: "This is somewhat unsurprising given the significant role social norms have been found to play in influencing how PEB is enacted in various circumstances (Bamberg & Möser, 2007; Heberlein, 2012; Thomas & Sharp, 2013). Despite the influence social norms have been found to have on PEB, they have not been the primary factor determining the structure of a widely used PEB scale previously". -> it is confusing why the topic of social norms appears here, may you clarify?

- In the Discussion (page 30) say: "Though this sub-scale structure has not been represented in previous studies, measuring general PEB through these two dimensions was determined to be psychometrically consistent across LNT and QUAL groups". -> this sentence is not clear because you say previously (in the same paragraph) that the bi-dimensional subscale structure is in line with previous research.

-In the Discussion (page 31) say: "For example, composting behavior was eventually removed from our scale due to its cross-loading between sub-scales amongst the QUAL group in the exploratory factor analysis,

indicating there may be some confounding factors influencing how this behavior is perceived and enacted. This aligns with findings by Huddart-Kennedy et al. (2009), suggesting that land stewardship behaviors are enacted less frequently in populations outside of rural areas". -> Here it is not clear whether you are talking about composting behavior or land stewardships behaviors, or ¿Are those behaviors being used interchangeably?

-In the Conclusion, you mention a scale with 11 final items as a tool for future researchers, however, I could not see this final list of PEB in your supplementary material. I think it may be helpful to know it.

-The final paragraph in the Conclusion is a bit confusing.

Reviewer #2: The paper addresses an important topic: How to measure pro-enviromental behavior in a rigorous way. This is very important for cumulative generation of knowledge, and I therefore like the endeavour very much. The authors also provide a very convincing analysis of what is currently missing, and what a scale that improves these issues would have to be like.

They then go on to develop such a scale in an overall very rigorous way. The final result is a scale with 11 items that works on a general population of the US and on a sample that is especially environmentally conscious.

I have a couple of points that I would like to see addressed by the authors.

First, the authors start with a pilot and then use the scale developed in this pilot on two samples, and perform both exploratory and confirmatory factor analyses on these two samples. It would have been nice to see actually a new sample to confirm the structure of the scale, however, this could also be done in a follow-up study. I would, however, like to see at least a discussion of this - why didn´t the authors go out-of-sample for the confirmatory factor analysis?

Secondly, the paper could be shortened considerably. The authors repeat their main points quite a lot, and by that prolong the text unneccesarily. Focusing the paper would increase its value and probably also the reception it will get by readers.

I am a bit surprised that the authors start with the argumentation that context very strongly influences PEB, but then go on to develop a scale that should be entirely consistent across two very distinct populations. Of course, having a very short scale is nice, and having only dimensions as well, in the sense of "ockham´s razor". However, I would like to see a more thorough discussion whether this does not under- or overestimate PEB in some subpopulations whose PEB is not captured in the items of the scale?

Finally, the authors mention that this study is part of an alternate study for which data were being collected. While I think publishing the scale-development of a larger project has merits, it would be important to have more information on the overall study.

A very minor point - in the description of the factor analyses, the authors mention that in the PAF they use a cutoff-value of 0.32, and later one of 0.30. Is there any argument for that difference?

6. PLOS authors have the option to publish the peer review history of their article (what does this mean?). If published, this will include your full peer review and any attached files.

Reviewer #1: **Yes: **Claudia Patricia Arias Puentes

Reviewer #2: No

---

## [Author Response · Author response to Decision Letter 0]

8 Jul 2022

Please see the attached reviewer response document for extensive details on the authors' response to the reviewers and Academic Editor.

---

## [Decision Letter · Decision Letter 1]

23 Aug 2022

A multi-dimensional measure of pro-environmental behavior for use across populations with varying levels of environmental involvement in the United States

PONE-D-21-27321R1

Dear Dr. Mateer,

We’re pleased to inform you that your manuscript has been judged scientifically suitable for publication and will be formally accepted for publication once it meets all outstanding technical requirements.

Kind regards,

Carlos Andres Trujillo, PhD

Academic Editor

PLOS ONE

Additional Editor Comments (optional):

Reviewers' comments:

Reviewer's Responses to Questions

**Comments to the Author**

1. If the authors have adequately addressed your comments raised in a previous round of review and you feel that this manuscript is now acceptable for publication, you may indicate that here to bypass the “Comments to the Author” section, enter your conflict of interest statement in the “Confidential to Editor” section, and submit your "Accept" recommendation.

Reviewer #1: All comments have been addressed

Reviewer #2: All comments have been addressed

2. Is the manuscript technically sound, and do the data support the conclusions?

Reviewer #1: Yes

Reviewer #2: Yes

3. Has the statistical analysis been performed appropriately and rigorously? 

Reviewer #1: Yes

Reviewer #2: Yes

4. Have the authors made all data underlying the findings in their manuscript fully available?

Reviewer #1: Yes

Reviewer #2: Yes

5. Is the manuscript presented in an intelligible fashion and written in standard English?

Reviewer #1: Yes

Reviewer #2: No

6. Review Comments to the Author

Reviewer #1: Dear authors, I found you answered most of my comments and broadly welcome the recommendations. Besides, the study's limitations were acknowledged. Now, you have a more helpful paper in the field of PEB.

Reviewer #2: (No Response)

7. PLOS authors have the option to publish the peer review history of their article (what does this mean?). If published, this will include your full peer review and any attached files.

Reviewer #1: No

Reviewer #2: No

---

## [Editor Report · Acceptance letter]

30 Aug 2022

PONE-D-21-27321R1 

A multi-dimensional measure of pro-environmental behavior for use across populations with varying levels of environmental involvement in the United States 

Dear Dr. Mateer:

I'm pleased to inform you that your manuscript has been deemed suitable for publication in PLOS ONE. Congratulations! Your manuscript is now with our production department. 

Kind regards, 

on behalf of

Dr. Carlos Andres Trujillo 

Academic Editor

PLOS ONE